# Morphofunctional Assessment beyond Malnutrition: Fat Mass Assessment in Adult Patients with Phenylketonuria—Systematic Review

**DOI:** 10.3390/nu16121833

**Published:** 2024-06-11

**Authors:** Luis M. Luengo-Pérez, Mercedes Fernández-Bueso, Carlos Guzmán-Carmona, Ana López-Navia, Claudia García-Lobato

**Affiliations:** 1Medical Sciences Department, Faculty of Medicine and Health Sciences, University of Extremadura, 06006 Badajoz, Spain; 2Clinical Nutrition and Dietetics Unit, Endocrinology and Nutrition Section, Badajoz University Hospital, 06008 Badajoz, Spain; mmercedes.fernandez@salud-juntaex.es (M.F.-B.); claudia.garcial@salud-juntaex.es (C.G.-L.); 3Endocrinology and Nutrition, Don Benito-Villanueva de la Serena Hospital Complex, 06400 Don Benito, Spain; carlos.guzman@salud-juntaex.es (C.G.-C.); ana.lopezn@salud-juntaex.es (A.L.-N.)

**Keywords:** phenylketonuria, body composition, body fat, morphofunctional assessment, cardiovascular risk, metabolic diseases

## Abstract

Morphofunctional assessment was developed to evaluate disease-related malnutrition. However, it can also be used to assess cardiometabolic risk, as excess adiposity increases this risk. Phenylketonuria (PKU) is the most prevalent inherited metabolic disease among adults, and obesity in PKU has recently gained interest, although fat mass correlates better with cardiometabolic risk than body mass index. In this systematic review, the objective was to assess whether adult patients with PKU have higher fat mass than healthy controls. Studies of adult PKU patients undergoing dietary treatment in a metabolic clinic reporting fat mass were included. The PubMed and EMBASE databases were searched. Relevance of articles, data collection, and risk of bias were evaluated by two independent reviewers. Ten articles were evaluated, six with a control group, including 310 subjects with PKU, 62 with mild hyperphenylalaninemia, and 157 controls. One study reported a significant and four a tendency towards an increased fat mass in all patients or only females with PKU. Limitations included not having a healthy control group, not reporting sex-specific results and using different techniques to assess fat mass. Evaluation of fat mass should be included in the morphofunctional assessment of cardiometabolic risk in adult patients with PKU.

## 1. Introduction

Morphofunctional assessment of patients’ nutritional status was first developed for disease-related malnutrition [1,2]. Malnutrition is an issue of concern in patients with inherited metabolic diseases (IMDs), some of whom have neurological manifestations, and most of whom are on restrictive dietary treatment.

Nevertheless, there are IMDs with a known increased cardiometabolic risk, such as homocystinuria and glycogen storage disease type III, while most patients with IMDs in adult metabolic clinics are patients with phenylketonuria (PKU), in whom a possible increased risk and prevalence of obesity have recently been discussed [3,4,5].

Newborn screening programs, together with dietary treatment, have led to an increase in the life expectancy of patients with PKU, with a growing number of patients aged 50 or older, in whom prevention of acquired cardiovascular and metabolic diseases should be considered, although it is not known whether the risk of cardiometabolic disease is similar or different to that of the general population.

Obesity, as an excess of total body adiposity, could increase the risk of acquired cardiometabolic disease. Adipose tissue has different biological endocrine, autocrine, and paracrine functions [6], and may increase the risk of cardiometabolic disease depending on its type and location [7,8].

In a global morphofunctional assessment of a patient’s nutritional status, fat mass should be evaluated as a morphological assessment of cardiometabolic risk. Excess adiposity and visceral fat are not well evaluated by body mass index (BMI) or other anthropometric methods. The Global Leadership Initiative on Malnutrition recommends anthropometry, only when technical approaches to assess muscle mass are not available [9]. Most available clinical fat assessment techniques are bioelectrical impedance analysis (BIA), and nutritional ultrasound, while others such as dual X-ray absorptiometry (DXA), computed tomography (CT), or magnetic resonance imaging (MRI) are less available or only opportunistic [10].

The objective of this systematic review was to evaluate whether patients with PKU have a higher cardiometabolic risk due to excess fat mass than people without hyperphenylalaninemia (hPA).

## 2. Materials and Methods

### 2.1. Protocol and Selection Criteria

This study was developed according to the Grading of Recommendations Assessment, Development and Evaluation (GRADE) systematic approach for evidence synthesis [11,12].

The inclusion and exclusion criteria for articles were defined following the PECO (Population, Exposure, Comparator, and Outcome) format [13].

Inclusion criteria were: Adult patients with PKU (population, exposure) undergoing dietary treatment at a metabolic diseases clinic (exposure); although published studies including a control group (comparator) were preferred, those without a control group were not excluded provided they were randomized controlled trials (RCTs), as well as non-randomized controlled trials (non-RCTs), or observational studies (case series, cohort, case-control, and cross-sectional), providing information about fat mass (outcome). Exclusion criteria: Pediatric population, patients with diseases other than PKU, lack of data on fat mass. The project has been registered at Open Science Framework (OSF): https://osf.io/f2xvn, accessed on 3 may 2024.

### 2.2. Search Strategy, Study Selection, and Data Collecction

A scientific literature search was performed on PubMed, and Embase databases on 21 September 2023 and updated on April 1st, 2024. Medical Subject Headings (MeSH) and text terms related to fat mass and PKU were used. Limits to “adult” patients and to the English, German, French, Portuguese, and Spanish languages were introduced. As the result in PubMed was only 9 articles, the “adult” limit was ignored at this stage of the review to obtain 33 articles (see Section 3.1).

All articles identified in the previous search were included in the screening process, but those with animals, seven more with pediatric populations only, two different studies with other inherited metabolic diseases and unrelated diseases, as well as duplicates, were subsequently excluded.

Two independent reviewers (L.M.L.-P. and C.G.-L.) assessed the relevance of the titles and abstracts of the articles. Full-text articles were reviewed when titles and abstracts did not provide enough information and were selected for their interest to assess eligibility according to the criteria in Section 2.1. As a secondary search strategy, the references included in the selected articles were screened to avoid missing relevant studies. There were some articles without a control group, with differences between reviewers, and it was decided to include them to avoid missing data, as the number of articles was low.

Data items from each study were extracted by two authors (L.M.L.-P. and M.F.-B.). Data included were: first author, country, year, study design, method, sample characteristics, including comparison with control group, and fat mass (outcome).

### 2.3. Assessment of Risk of Bias in Individual Studies

Two reviewers (L.M.L.-P. and M.F.-B.) independently evaluated the risk of bias of selected studies using the National Institutes of Health (NIH) Quality Assessment Tool for Observational Cohort and Cross-Sectional Studies [14], including the items related to: Statement of the research question or objective, definition of the study population, rate of eligible persons, inclusion and exclusion criteria, sample size justification, when and how exposure and outcome were measured, blinding of outcome assessors, rate of follow-up, and adjustment of confounding variables. Disagreements between reviewers were resolved by consensus, with the highest risk of bias identified when there were differences.

## 3. Results

### 3.1. Study Selection

A total of 149 articles were identified in EMBASE and 33 in PubMed (without “Adult” limit). Figure 1 shows the flow diagram of the review process following the PRISMA model [15].

After removing 13 duplicates, the titles and abstracts of the remaining 169 articles were assessed for relevance, and a total of 38 full-text articles were selected for evaluation.

Among these 38 articles, 22 did not provide information about body composition, 5 included an exclusively or predominantly pediatric sample, and another one did not include patients with PKU but healthy volunteers. (See Appendix A). The remaining 10 articles fulfilled all inclusion criteria (Section 2.1) and were included for quantitative analysis [16,17,18,19,20,21,22,23,24,25].

### 3.2. Study Characteristics

A summary of the 10 articles reviewed is shown in Table 1.

There was only one longitudinal study [20], with a follow-up of six months. The remaining 9 studies were cross-sectional, although one of them [23] was a cross-sectional study of a subset of 15 participants from a previously published randomized, crossover trial [26], and another one included baseline data from a longitudinal study [18].

Five studies were developed in Europe [16,17,20,21,25], four in America [18,19,22,23], and one in Asia [24]. All of them, but Rocha et al. (2012) [21], were published between 2017 and 2023.

Three studies did not include a control group [18,20,23], and Jani et al. [18] compared patients with PKU with the USA reference population. These three studies included 46 patients with PKU. The other six articles included 168 patients with PKU and 157 non-HPA-matched controls [16,17,19,21,22,24], and the last study, by Zerjav Tansek et al. [25], compared outcomes in 96 patients with PKU and 62 patients with mild HPA. Considering all the articles, outcomes in 310 subjects with PKU, 62 with mild HPA, and 157 with non-HPA were communicated.

All studies included both female and male participants (patients with PKU and controls), but two of them did not report the number of each [20,22], and the other four did not report female and male results separately (Table 2) [16,21,24,25].

Most articles included a mixed sample of pediatric and adult patients (and controls). Whenever adult data were presented separately [16,18,20,21], only these data were taken into consideration. Rocha et al. [21] included as adult patients those aged 16 or older.

Two other studies with mixed pediatric-adult samples, reported the proportion of adult patients [25] or did not report [24], but neither of them reported adult data separately. The remaining 3 studies included only [17,19] or mainly [23] adult patients, as Stroup et al. included 3 patients aged 15 to 17 among their sample of 15 patients.

All the included studies assessed body composition, as fat mass was the PECO outcome inclusion criteria of this review (Section 2.1). Body composition was assessed by BIA [17,19,21,24] and DXA [18,22,23,25] in four articles each, and plethysmography [20] and deuterium [16] were the methods used in the remaining two studies.

Two studies [20,23] did not compare fat mass in patients with PKU with any other group. Among the six articles that included patients with PKU and non-HPA-matched controls [16,17,19,21,22,24], only Barta et al. [17] found that adult female patients with PKU had significantly more fat mass than non-HPA matched controls. Jani et al. [18] found a similar outcome comparing patients with PKU to the US reference population, but significance was not provided. Zerjav Tansek et al. [25] found that only patients with classic PKU had significantly lower fat mass than patients with non-classic PKU, but they included data from the pediatric population, with no other differences in whole-body or abdominal fat mass.

Fat mass from cross-sectional studies is shown in Table 2 (data reported without differences by sex)and Table 3 (male and female data separately).

### 3.3. Risk of Bias Assessment

Two of the studies [19,21] were fair, with a moderate risk of bias, and the remaining eight [16,17,18,20,22,23,24,25] were poor, with high risk of bias. Figure 2 shows the degree of compliance with the selected items (see Section 2.3) from the NIH Quality Assessment Tool for Observational Cohort and Cross-Sectional Studies [14]. Risk of bias analysis from every individual study is available in the Appendix A.

### 3.4. Synthesis of Results

#### 3.4.1. Patients with PKU vs. Controls

Six of the studies [16,17,19,21,22,24] compared fat mass in patients with PKU with non-HPA controls, and only one of them, by Barta et al. [17], found significantly higher fat mass in adult female patients with PKU compared with matched healthy controls (35.8% vs. 24.7%, *p* = 0.028). No other significant differences in fat mass were found.

Zerjav Tansek et al. [25] found that the proportion of whole-body fat by DXA was significantly (*p* = 0.04) lower in patients with classic PKU (24.3% ± 6.4) compared to non-classic PKU (27.6% ± 6.9), but they did not compare with healthy volunteers.

#### 3.4.2. Patients with PKU without Control Group

Three studies did not include a control group [18,20,23].

Jani et al. [18] only showed that the median fat mass index by DXA in 26 adult patients with PKU (mainly females) was 9.1 (range: 5.3–29.5) and the median fat mass in males was 30.9 kg (17.1–51.4) and 23.4 kg (13.8–81.4) in females, and compared these results with the median fat mass in the US reference population, 23.7 and 28.7 kg, respectively.

Stroup et al. [23] found fat mass by DXA in 6 males with PKU was 24.5% ± 4.8, and, in 9 females, 36.5% ± 2.5, but included 20% of subjects aged less than 18.

Montanari et al. [20] found a median basal fat mass of 15.2% by plethysmography, which increased significantly after 6 months of study to 19.6%, and which increased from the 25th to the 50th percentile in females but did not change in males.

#### 3.4.3. Metabolic Control

In the study by Alghamdi et al. [16], all adults with PKU had poor metabolic control, as 100% of phenylalanine measurements were above age-specific targets.

In the sample of Barta et al. [17], poor metabolic control of PKU, defined as mean Phe level > 600 μmol/L over the course of 10 years, was observed in 35% and 67% of male and female patients with PKU, respectively, and there was no significant inverse correlation between amino acid supplementation intake and fat mass.

Mezzomo et al. [19] observed good metabolic control of PKU in 41.67% (15/36) of the sample, being 65% (13/20) in males and 12.5% (2/16) in females, but they did not present the results of fat mass related to metabolic control, although the authors said, “The individuals with PKU evaluated here are young adults with inadequate metabolic control of the disease, with males being eutrophic and females being overweight and excess BF”.

Montanari et al. [20] found that metabolic control of PKU was worse (not significantly) at the end of the study.

Rocha et al. [21] observed that 40.5% of their patients older than 16 years had good metabolic control of PKU. The authors found that the prevalence of overweight was higher in patients with poor metabolic control, but they did not report fat mass with respect to metabolic control.

Rojas Agurto et al. [22] compared a group of patients with PKU under strict follow-up with another group of patients who stopped attending metabolic control visits, but they found no significant differences in their body fat mass.

The remaining articles [18,23,24,25] did not provide information regarding metabolic control of PKU.

As information on metabolic control in PKU is scarce and heterogeneous, it was not possible to explore the association between metabolic control and fat mass.

#### 3.4.4. Sex

Five of the studies did not show differentiated adult fat mass in females and males [16,21,22,24,25], although Rocha et al. found a non-significantly higher fat mass in females with PKU, including children (27.5%) than in healthy controls (26.1%, *p* = 0.192).

Among the five ones that did show these data, Barta et al. [17] found that adult female patients with PKU had significantly more fat mass than control females. Mezzomo et al. [19] found no significant differences. Jani et al. [18] did not compare with a control group but with a US reference population. Montanari et al. [20] found that female patients with PKU increased their fat mass at the end of the study. Stroup et al. [23] did not compare fat mass in PKU patients with a control group.

#### 3.4.5. Body Fat Mass

Fat mass was assessed by DXA in 4 studies [18,22,23,25], BIA in another 4 [17,19,21,24], and by plethysmography [20] or deuterium [16] in one study each. This heterogeneity made it unfeasible to compare the body fat mass outcome.

#### 3.4.6. Moderate vs. Poor Risk of Bias Studies

Only two of the studies [19,21] were considered fair, with a moderate risk of bias, and these found no significant differences in fat mass between patients with PKU and healthy controls. Among the studies with a high risk of bias [16,17,18,22,23,24,25], Barta et al. [17] found a significantly higher fat mass in female patients with PKU compared to healthy controls.

## 4. Discussion

Morphofunctional assessment of patients’ nutritional status was first developed for disease-related malnutrition [1,2]. Malnutrition is an issue of concern in patients with IMDs with severe neurological complications, but there are IMDs with known increased cardiometabolic risk, and most patients with IMDs in adult metabolic clinics are patients with PKU, in whom a possible risk and prevalence of obesity has recently been discussed [3,4,5], as obesity could increase acquired cardiometabolic risk.

### 4.1. Inherited Metabolic Diseases with Known Higher Cardiovascular Risk

Homocystinuria is an IMD with a well-characterized acquired increased cardiometabolic risk. Although McCully and Wilson proposed the “Homocysteine Theory of Atherosclerosis” [27] in 1975, it was not until the 1990s that increased serum levels of total homocysteine were recognized as a new independent risk factor for cardiovascular disease [28].

A direct association has been found between increased plasma total homocysteine levels and cardiovascular disease [29]. Hyperhomocysteinemia leads to:vascular endothelial injury and dysfunction, with less release of nitric oxide, thus favoring endothelial dysfunction and the atherothrombotic process [30];proliferation of vascular wall smooth muscle cells [31]; lipid peroxidation, with oxidation of LDL-C through the generation of the superoxide radical [32];a prothrombotic state favored by an increase in the activity of coagulation factors V and XII and a higher production of thromboxane A2 (a potent platelet aggregator), favoring the genesis of vascular disease [33];intraluminal venous thrombi formation [34].

Cardiovascular risk in homocystinuria is expounded in the result of two meta-analyses of case-control and prospective studies, indicating that for every 5 υmol/L of increased homocysteine, the risk of ischemic heart disease increases by 56.8%, and the risk of stroke by 61.3% [35,36].

Another IMD with a known increased cardiometabolic risk is glycogen storage disease type 1, which induces hyperlactacidemia, hyperlipidemia, and hyperuricemia [37]. The marked hyperlipidemia may be due to increased de novo lipidogenesis and release of lipids into the blood compartment [38], and decreased clearance and uptake, on account of reduced activity of lipoprotein lipase and hepatic lipase [39]. Arterial dysfunction is also present, leading to an increased cardiovascular risk due to hyperlipemia [40].

Most IMDs do not lead per se to a higher cardiometabolic risk than in the general population, but this risk could be increased in patients with excess adiposity.

### 4.2. Adipose Tissue and Cardiometabolic Risk

Adipose tissue (AT) is a complex and dynamic endocrine organ. Its biological variability, depending on its location and metabolic state, affects individuals and their cardiometabolic risk [7,8].

There are two major types of adipose tissue in the body (classified by phenotype and functional role): White adipose tissue (WAT) and brown adipose tissue (BAT). WAT can be found in two main anatomical depots: Ectopic or visceral adipose tissue (VAT), which is strongly associated with cardiometabolic risk, and subcutaneous adipose tissue (SAT) [7,8]. VAT has higher levels of macrophages, regulatory T cells, natural killer T cells, and eosinophils than SAT [7], and both display differences in angiogenesis and sympathetic inervation [41].

ATs are known to have endocrine, autocrine, and paracrine functions [6].

In obesity, the secretion of hormones and adipokines is different compared to normal weight individuals [42,43,44,45]. This results in an increased risk of coronary artery calcification, carotid artery intimal media thickening, and left ventricular hypertrophy [44,45,46].

Adyponectin is reduced in obesity, increasing the risk of hypertension, myocardial hypertrophy, and endothelial dysfunction; omentin, SPARC, and nefastin-1 levels, which are inversely correlated with cardiovascular disease, are also reduced in obesity. Hyperleptinemia is associated with cardiovascular remodeling, prothrombotic effects, and endothelial dysfunction; resistin, angiotensinogen, and visfatin are also expressed in VAT, increased in obesity, and related to high blood pressure; chemerin is associated with inflammatory markers in metabolic syndrome; other adipokines such as lipocalin-2 (LCN2), vaspin, FSTL1, SFRP5, CTRPs, FAM19A5, WISP1,PGRN, apelin, RBP4, PAI-1 are also secreted by AT, and their levels have a correlation with major risk of CV disease [7,43].

Cardiovascular disease (CVD) is a leading cause of morbidity and mortality worldwide. Obesity, or more precisely, increased ectopic adiposity, is associated with cardiometabolic abnormalities such as hypertension, dyslipidemia, and insulin resistance [7,8]. Fat mass distribution should be considered the main indicator of CV risk, and therefore fat mass must be evaluated in the morphofunctional assessment of cardiometabolic risk.

### 4.3. Morphofunctional Assessment of Cardiometabolic Risk

Visceral fat can be assessed by means of anthropometric measurements. The most commonly used marker [47], body mass index (BMI), should not be useful [42,47], as its main limitation is the inability to discern the composition and distribution of fat mass from muscle mass. Therefore, other anthropometric markers focused on the measurement of central fat mass, which are more closely related to cardiometabolic risk, are also employed [48].

Waist circumference (WC) is the anthropometric marker that best predicts intra-abdominal fat mass [49,50]. A waist circumference > 102 cm in men and >88 cm in women is considered a marker of cardiometabolic risk [51]. Other anthropometric measurements that can estimate visceral fat are waist-to-height ratio [52], waist-to-hip ratio [52], and relative fat mass, which considers WC, height, and sex to express total body fat as a percentage and shows a stronger correlation between total body fat and BMI [53,54].

Visceral adipose index is a mixed morphological and functional parameter. It is based on WC and BMI, as well as triglycerides and HDL cholesterol, and shows a strong association with visceral adipose tissue composition measured by MRI. It also indirectly expresses visceral adipose tissue function and insulin sensitivity by using different formulae according to sex and ethnicity [55,56].

Anthropometric techniques are imprecise for evaluating visceral/ectopic fat mass, and more specific evaluation methods are available [1,10,57].

**Bioelectrical impedance analysis** (BIA) performs an estimation of body composition, including fat mass, by measuring the resistance of body tissues to electrical currents. These currents are of low intensity and high frequency [58]. The main limitation of BIA when assessing adipose tissue, in addition to the limitations of its own estimation, is the hydration variation, as BIA estimates FFM assuming that FFM is constantly hydrated to 73.2% [59]. FFM hydration status is also higher in individuals with obesity; therefore, total body composition estimates are affected [59,60].

**Nutritional Ultrasound**^®^ (NU) is a tool capable of evaluating adipose tissue (visceral and muscular ectopic adipose tissues). The distribution of ectopic adipose tissue is represented by preperitoneal visceral fat (PVF), which is related to other adipose tissue deposits as intrahepatic fat. There are limitations in PVF measurement in persons with obesity due to the impossibility of covering the layer depth with nutritional ultrasound [61]. Ectopic adipose infiltration of muscle is a type of ectopic adipose tissue that is known as myosteatosis. NU evaluates muscular ectopic adipose tissue by measuring echointensity, as fat is usually more echolucent and superficially distributed [62,63]. Echointensity can be assessed from two points of view. Qualitatively, according to hypo, iso, or hyperintensity; quantitatively, using grayscale analysis on ultrasound with different available software [64]. Furthermore, color Doppler ultrasound can be used to observe quadricep rectus femoris vascularization. Muscle fat infiltration involves decreased vascular flow [65].

Due to radiation exposure, **computed tomography** (CT) is generally not used as a first choice for body composition measurement, but rather to evaluate the body composition of CTs previously performed for another purpose. Tissue is measured using a quantitative scale in Hounsfield units (HU), a measure of body tissue attenuation compared to water [16]. Muscle mass, visceral, subcutaneous, pericardial, intermuscular, and intramuscular adipose tissue can be identified based on predetermined HU values [58,66]. For muscle adipose tissue assessment, either intermuscular adipose tissue (IMAT) or muscle attenuation can be measured [67]. Low muscle attenuation is associated with increased muscle fat content as well as decreased specific strength [68]. The area of the different tissues can be determined manually or through software, with a typical range attributed to fat, from −195 to −45 HU [58,66,69].

**Magnetic resonance imaging** (MRI), compared to CT, provides a better definition of soft tissues, especially fat [70]. MRI is a useful tool for measuring muscle mass and subcutaneous, visceral, and ectopic adipose tissue [71] and shows a high correlation with CT [72]. However, MRI is not a first-choice technique for body composition assessment due to its higher cost and lower availability.

**Dual-energy X-ray absorptiometry** (DXA) performs a full-body scan using low doses of radiation in a short period of time, with high precision and accuracy. DXA indirectly measures lean mass and fat mass. DXA has a good correlation with BIA, CT, and MRI, although CT and MRI are more accurate in the assessment of visceral fat and ectopic adipose tissue, and DXA cannot measure intramuscular fat mass [58].

CT, MRI, and DXA are usually less available in clinics than BIA and NU. Other techniques to evaluate ectopic fat are more experimental, such as plethysmography and deuterium dilution, but they have also been included in the systematic review.

In the **functional assessment** of cardiometabolic risk, it should be taken into account that adipokines produced in the adipose tissue are key to insulin resistance, inflammation, and tissue dysfunction [73]. Ectopic/visceral adipose tissue is characterized by tissue infiltration by macrophages and leukocytes and the secretion of inflammatory cytokines, including *C*-reactive protein, IL-6, plasminogen activator inhibitor-1 (PAI-1) and TNF-α [74,75]. Ectopic adipose tissue also modifies the plasma lipid profile through increased lipolysis and the release of free fatty acids, decreased expression of lipoprotein lipase (LPL), and increased expression of cholesterol ester transfer protein (CETP). This leads to a lipid profile characterized by hypertriglyceridemia, increased LDL and low-density lipoproteins, and low HDL cholesterol levels [75].

For the consequences of ectopic fat deposition, the functional assessment of cardiometabolic risk should include blood pressure, glucose (HOMA-IR), lipid metabolism (HDL-cholesterol, triglycerides, and their ratio), and inflammatory markers (PCR).

This systematic review focuses on morphological fat assessment in patients with PKU, the most prevalent IMD in adult metabolic clinics [76].

### 4.4. Summary of Evidence

The association between PKU and obesity had been previously explored in three systematic reviews, one of which included a meta-analysis. The first one, in children and adolescents only, found that overweight was a common outcome in this population [77], the second one, with adults and pediatric populations with PKU, found no differences in BMI between patients with PKU and healthy controls, but patients with classical PKU had a significantly higher BMI than healthy controls [4], and the last one, only in adults, found patients with PKU had a higher BMI but also a higher prevalence of obesity than healthy controls, but with inconsistent results when compared with the general population [5].

Nevertheless, no published systematic review was found on the assessment of fat mass in patients with PKU, despite the fact that fat mass (namely ectopic fat mass) correlated better with cardiometabolic risk than BMI.

One six-month longitudinal follow-up study [20] and nine observational studies [16,17,18,19,21,22,23,24,25] were reviewed, including a cross-sectional analysis of two other studies [18], and a cross-sectional study [23] of a subset of 15 participants from a previously published randomized crossover trial [26].

Only six of the studies compared results with a non-hPA control group [16,17,19,21,22,24], while the longitudinal study [20] and the remaining three observational studies did not [18,23,25]. Only Barta et al. [17] found a significantly higher fat mass in female patients with PKU, although Alghamdi et al. [16], and Weng et al. (including children) [24] found a non-significantly higher fat mass in patients with PKU versus non-hPA controls, while Mezzomo et al. [19], and Rocha et al. [21] reported a higher non-significant fat mass in female patients with PKU vs. non-hPA females.

In the only longitudinal study included [20], there was a significant increase in fat mass percentile in female patients with PKU, but not in males, although all of them were in a 6-month follow-up interventional study with a GMP formula. This may reflect how fast fat mass can fluctuate over time, at least in women, and it is related to the results from cross-sectional studies, which reflect a tendency towards higher fat mass in female PKU patients than in females in the general population.

As body composition changes with age, it is necessary that outcomes are only studied in adult samples or, at least, reported separately for pediatric patients and controls. Four of the selected articles [16,18,20,21] included both pediatric and adult patients, but they were included because they reported outcome data separately for adults [16,18,20] or patients older than 16 years [21] without increasing the risk of bias. Stroup et al. [23] included three patients (20%) aged 15 or older. The study with the highest risk of bias due to the inclusion of data from pediatric patients is that of Weng et al. [24], as they did not report the number of adults in the PKU or control groups, nor did they report outcome data for adults separately.

Among these six studies that compared results in patients with PKU with a control group, the ones from Mezzomo et al. [19] and Rocha et al. [21] were evaluated with moderate risk of bias, according to the NIH Quality Assessment Tool for Observational Cohort and Cross-Sectional Studies [14], and reported a tendency towards higher fat mass among female patients with PKU (Rocha et al., including pediatric patients). The remaining 4 studies, assessed as high risk of bias, reported no tendency [22], tendency towards [16,24] or significantly higher [17] fat mass in patients with PKU.

Exposure was defined as PKU under treatment, but different levels of exposure (compliance, BH4 treatment, etc.) were not reported in most cases. Good metabolic control in patients with PKU was defined as the Phe concentration in a single sample [18], the median [21] or average [22] concentration in the previous year or the preceding 10 years [17], attendance at follow-up appointments [22], and regular intake of protein substitutes without Phe were also considered [17,22].

Barta et al. did not find any significant correlation between metabolic control and body composition [17]. Rojas Agurto et al. [22] reported that patients with PKU and poor metabolic control had a fat mass of 24.4 (18.1–32.4) kg compared to 21.1 (16–30.2) kg in patients with good metabolic control, but did not show statistical analysis. Rocha et al. found that patients with poor metabolic control had a prevalence of overweight and obesity of 42.9%, compared with 27.9% in patients with good metabolic control, but they did not report the relationship between fat mass and metabolic control [21].

Fat mass was not adjusted for other variables related to fat mass deposition, such as energy intake, physical activity, sociocultural status, or family history.

The quality of the evidence was very low due to the risk of bias in the studies reviewed, but it may be enough to raise the possibility of higher fat mass, and subsequently, an increased cardiometabolic risk in people with PKU.

In the context of morphofunctional assessment of patients with PKU, body fat mass and, when available, ectopic fat mass should be included in the morphological evaluation as biomarkers for the early detection of cardiometabolic risk, followed by functional evaluation including HOMA-IR, triglycerides, HDL-cholesterol, LDL-cholesterol, and PCR. Protein substitutes need to be designed in order to avoid insulin resistance, dyslipidemia, and systemic inflammation.

### 4.5. Strengths and Limitations of This Study

There are some limitations to this systematic review. Just 10 studies were evaluated, after rejecting 28 (see Appendix A) [4,26,78,79,80,81,82,83,84,85,86,87,88,89,90,91,92,93,94,95,96,97,98,99,100,101,102,103], and only six of which included a matched control group, and all these six studies were observational, which have a higher risk of bias due to confounding variables than randomized clinical trials.

The populations included in the studies were diverse in terms of age, disease severity, type of treatment and adherence, and metabolic control. Zerjav Tansek et al. [25] compared fat mass in patients with PKU with HPA as controls and in patients with classic PKU compared to non-classic PKU. In this study, 50% of the patients with PKU but only 13% of benign HPA patients were adults, and the proportion of females was similar in both groups (51–52%), but outcomes from adults vs. non-adults and females vs. males were not differentiated, despite the fact that the fat mass in females is higher and changes with age, both physiologically.

Five of the studies did not report separately fat mass in adult females and males [16,21,22,24,25], although fat mass is physiologically different according to sex and should be reported properly. There are five studies from Europe [16,17,20,21,25], two from the USA, two from South America, and one from Asia. As there are differences in body composition according to ethnicity, it is difficult to compare the results.

There are also differences in the metabolic control of patients with PKU, with the proportion of patients in good metabolic control ranging from 65% in males in Mezzomo et al. [19] to 0% in the study by Alghamdi et al. [16], as all phenylalanine measurements were above age-specific targets.

The methods employed for fat mass assessment were also different: DXA in four [18,22,23,25] and BIA in another four [17,19,21,24]. Alghamdi et al. [16] assessed fat mass by deuterium but did not report sex-specific results. Montanari et al. [20] measured fat mass with plethysmography in 4 adults (45% of subjects); fat mass was not a primary objective in this study, and they do not show adult data separately.

There were also different ways of presenting the fat mass outcome. Jani et al. [18] presented fat mass outcome as a fat mass index and distinguished between adult and pediatric populations, but not between females and males. They also presented median fat mass in kilograms, as Rojas Agurto et al. [22], which can be influenced by height, and so fat mass index or percentage could be better alternatives. The only study reporting ectopic fat mass (abdominal fat mass), which correlates better with cardiometabolic risk, is the one from Zerjav Tansek et al. [25], but they did not report female abdominal fat mass results apart from male results.

The strength of the evidence was very low because only two of the trials had a moderate risk of bias and the other eight had a high risk, resulting in a limited quality of evidence for the body fat-mass percentage. Furthermore, differences from the technique employed to evaluate fat mass, absence of control group in four articles [18,20,23,25], absence of separated data from females and males [16,18,21,22,23,24], and age of patients and controls included, were not adjusted for and it not possible to conduct a meta-analysis.

To our knowledge, this is the first systematic review regarding fat mass assessment in patients with PKU. The following methodology was used in this systematic review:Followed the PRISMA guidelinesClearly defined the objective of this reviewDefined inclusion and exclusion criteria according to the PECO formatIncluded both PubMed and EMBASE databases in the search strategyPresented the full search strategies for both databases, including any filters and limits usedSearched the reference lists of the included studiesDescribed the study selection process using the PRISMA-model flow diagramProvided the list of excluded studies and the reasons for their exclusion in the Appendix AProvided a table with the main characteristics of the included studiesStudy selection, data search, and assessment of risk of bias and quality of evidence were performed by two independent authorsDescribed the rationale for the review in the context of existing knowledgeProvided an interpretation of the results in the context of the evidenceDiscussed the limitations of the evidence included in the review and the limitations of the review process itself.

This systematic review provides an updated overview of the evidence on fat mass in patients with PKU and may be useful to assess body and ectopic fat mass as a part of morphofunctional assessment and to include cardiometabolic disease assessment, prevention, and follow-up programs for patients with PKU in adult IMD clinics.

Morphofunctional assessment of cardiometabolic risk should be complemented by a functional evaluation, which is proposed to include not only known major cardiovascular risk factors (dyslipidemia, high blood pressure, diabetes mellitus, and smoking) but also known markers of these conditions (HOMA-IR) or systemic inflammation (PCR).

## 5. Conclusions

Fat mass (namely ectopic fat mass) correlates with cardiometabolic risk better than BMI, and therefore fat mass should be assessed preferentially, as a global morphofunctional assessment of nutritional state should include not only disease-related malnutrition but also cardiometabolic risk.

There were no significant differences in fat mass between adult patients with PKU and healthy controls, although a tendency towards higher fat mass in patients, particularly among women, was shown.

The quality of the evidence was very low due to the risk of bias in the studies reviewed, but it may be enough to raise the possibility of an increased cardiometabolic risk in people with PKU.

As a second part of morphofunctional assessment, after morphological fat mass assessment, functional evaluation of dyslipidemia, insulin resistance, and systemic inflammation should be performed.

More evidence is needed on body composition, body and ectopic fat mass, and cardiometabolic risk in patients with PKU.

## Figures and Tables

**Figure 1 nutrients-16-01833-f001:**
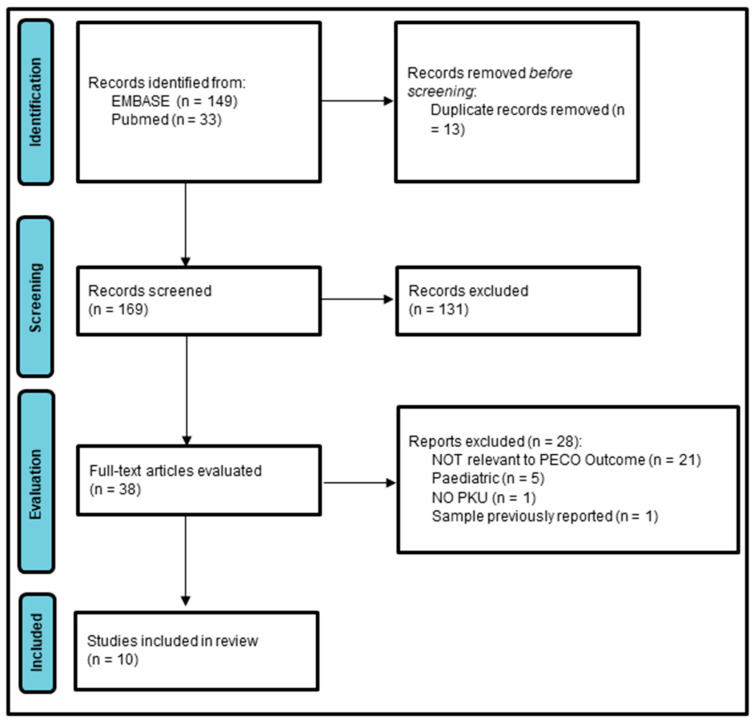
Study flow diagram following PRISMA model.

**Figure 2 nutrients-16-01833-f002:**
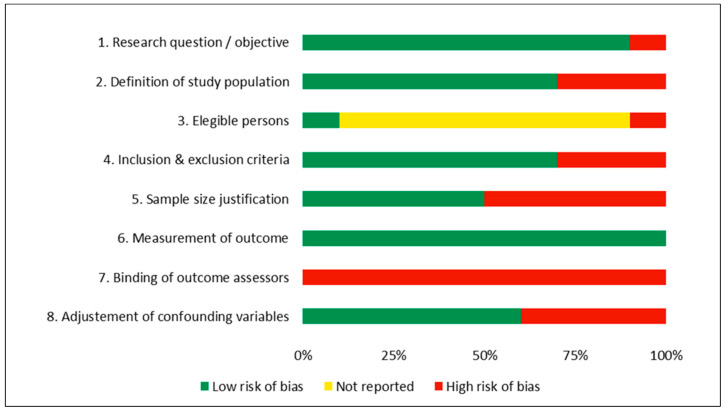
Risk of bias: judgments about each risk of bias item presented as percentages across all included studies.

**Table 1 nutrients-16-01833-t001:** Summary of the studies included in the systematic review.

Reference (Country)	Study Design (Duration of Follow-Up)	Sample Size (Age)	Controls (Age)	Sex (F/M)	Other	Risk of Bias ^1^
Alghamdi et al. 2021 (UK) [16]	Cross-sectional	10 (33.9 ± 5.0)	9 (28.8 ± 5.9)	P: 6/4 C: 6/3	Mixed pediatric and adult sample	High
Barta et al. 2022 (Hungary) [17]	Cross-sectional	50 (F 31 ± 7.8, M 26.6 ± 7.6)	40 (F 26.5, M 24)	P: 27/23 C: 20/20	-	High
Jani et al. 2017 (USA) [18]	Cross-sectional	27 (28.8, [19.5–54.6])	NO	18/9	Mixed pediatric and adult sample, compared with reference US population	High
Mezzomo et al. 2023 (Brazil) [19]	Cross-sectional	36 (25.36 ± 5.14)	33 (28.27 ± 6.15)	P: 16/20 C: 21/12	-	Moderate
Montanari et al. 2022 (Italy) [20]	Longitudinal (6 months)	4 (n.a.)	NO	n.a.	Mixed pediatric and adult sample	High
Rocha et al. 2012 (Portugal) [21]	Cross-sectional	26 (22.8 ± 3.0)	29 (23.6 ± 4.7)	n.a.	Mixed pediatric and adult sample	Moderate
Rojas Agurto et al. 2023 (Chile) [22]	Cross-sectional	24 (39.3)	24 (38.4)	P: 10/14 C: 10/14		High
Stroup et al. 2018 (USA) [23]	Cross-sectional	15 (15–50)	NO	9/6	Included 3 adolescents (15–17 y)	High
Weng et al. 2020 (Taiwan) [24]	Cross-sectional	22 (15.23 ± 5.23 [8–27])	22 (19.73 ± 10.6 [8–39])	P: 12/10 C: 12/10	Correlates inversely with protein intake Adult subjects number not shown	High
Zerjav Tansek et al. 2020 (Slovenia) [25]	Cross-sectional	96 (48 adults) (22.2 ± 11.4)	NO/62 mild HPA (14.4 ± 6.8)	P: 50/46 HPA: 22/40	Compared with mild HPA, not healthy controls	High

^1^ Assessed using the National Institutes of Health (NIH) Quality Assessment Tool for Observational Cohort and Cross-Sectional Studies [14]. Abbreviations: AFM: abdominal fat mass, C: control group, F: female, FM: fat mass, FMI: fat mass index, HPA: hyperphenylalaninemia, M: male, n.a.: not available, P: PKU patients group, UK: United Kingdom; USA: United States of America. n.b.: When samples were with mixed pediatric and adult populations, only adult (or older than 15/16) results are shown.

**Table 2 nutrients-16-01833-t002:** Fat mass outcome from references that do NOT report adult female and male fat mass separately.

Reference (Country)	Parameter, Technique	PKU	Control	Difference	*p*
Alghamdi et al. 2021 (UK) [16]	FM (%) FMI Deuterium	39.4 ± 8.2 12.9 ± 4.6	34.3 ± 11.1 11.0 ± 5.8	+5.1 +1.8	n.a. n.a.
Rocha et al. 2012 (Portugal) [21]	FM (%), BIA	23.8 (13.9, 35.5)	23.8 (17.9, 34.3)	0	0.964
Rojas Agurto et al. 2023 (Chile) [22]	FM (kg), DXA	23.15	24.56	−1.41	n.a.
Weng et al. 2020 (Taiwan) [24]	FM (%), BIA	20.74 ± 8.9	18.67 ± 7.52	+2.07	0.4635
Zerjav Tansek et al. 2020 (Slovenia) [25]	FM (%) AFM (%) DXA	25.8 ± 6.8 22.7 ± 7.8	25.4 ± 6.7 21.1 ± 7.2 (HPA)	+0.4 +1.6	0.758 0.204

Data reported as mean ± standard deviation or mean (P25, P75). Abbreviations: AFM: abdominal fat mass, BIA: bioelectrical impedance analysis, DXA: dual X-ray absorptiometry, FM: fat mass, FMI: fat mass index, HPA: hyperphenylalaninemia, n.a.: not available, UK: United Kingdom. n.b.: When samples were with mixed pediatric and adult populations, only adult (or older than 15/16) results are shown.

**Table 3 nutrients-16-01833-t003:** Fat mass outcome from references that report adult female and male fat mass separately.

Reference (Country)	Parameter, Technique	Female PKU	Female Control	Difference	Male PKU	Male Control	Difference
Barta et al. 2022 (Hungary) [17]	FM (%), BIA	36.7 (30.6, 40.2)	24.7 (22.2, 30.8)	+12.0 *****	18.7 (14.3, 29.8)	19.4 (15.07, 24.5)	−0.7
Jani et al. 2017 (USA) [18]	FMI, DXA	38.9 ******* (30.8, 64.3)	40.7 ******	−1.8	23.4 ******* (13.8, 81.4)	28.7 ******	+5.3
Mezzomo et al. 2023 (Brazil) [19]	FM (%), BIA	36.2 (20.1, 49.0)	28.4 (15.9, 46.4)	+7.2	17.4 (10.1, 29.5)	23.3 (12.1, 27.2)	−5.9
Stroup et al. 2018 (USA) [23]	FM (%), DXA	36.5 ± 2.5	-	-	24.5 ± 4.8	-	-

***** *p* = 0.078; all the rest, *p* > 0.05. ****** USA reference population as control group. Data reported as mean ± standard deviation or mean (P25, P75) or ******* mean (max, min). Abbreviations: BIA: bioelectrical impedance analysis, DXA: dual X-ray absorptiometry, FM: fat mass, FMI: fat mass index, USA: United States of America. n.b.: When samples were with mixed pediatric and adult populations, only adult (or older than 15/16) results are shown.

## Data Availability

Data available at:
Review fat mass PKU.

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
