# Peer review of "Morphofunctional Assessment beyond Malnutrition: Fat Mass Assessment in Adult Patients with Phenylketonuria—Systematic Review"

_nutrients, 2024, doi:10.3390/nu16121833_

Round 1
Reviewer 1 Report
Comments and Suggestions for Authors
To the Authors,
I am pleased to realize the peer review of the systematic review entitled "Morphofunctional assessment beyond malnutrition: fat mass assessment in adult patients with phenylketonuria. Systematic review" submitted to the Nutrients Journal. The aim of this research is to evaluate the cardiovascular risk of adult patients with PKU compared to those without this metabolic disease. The introduction provides a comprehensive background on the limitations of the current anthropometric methods used to evaluate the distribution of adipose tissue and its relevance for cardiometabolic risk. The methods are clearly presented and provide sufficient information to present the design of this study. In the results section, the presentation of the studies included in the analysis is sound and well -structured, so that the material is easy to follow. The authors identified the major methods used to evaluate the adipose tissue in PKU studies and the relation between AT and PKU metabolic control in studies with and without control group. The risk of bias evaluation in the studies included in the review is clearly depicted.
Major revisions
1. My suggestion is to provide the rationale to evaluate the cardiovascular risk in patients with PKU in relation to their life expectancy or the prevalence of cardiovascular diseases as compared to people without PKU in the introduction. The gap in knowledge in this area should be identified, which would provide a more solid motivation for this review.
2. The systematic review identified adipose tissue proportion and distribution as a cardiovascular risk factor. However, other risk factors that are present in the general population were presented in relation to other inherited metabolic diseases but were not included in the analysis in patients with PKU. The presence of dyslipidemia, high blood pressure, age, sex, and smoking are major risk factors that are used in the SCORE diagrams for the evaluation of cardiovascular risk in people without inherent metabolic diseases. Diabetes and prediabetes are also major cardiovascular risk factors. The presence of these factors in the studies addressing patients with PKU should also be presented. If possible, it would be useful to provide a possible explanation for the reason why these factors were not evaluated in the studies included in the analysis.
Sincerely yours
Author Response
Answers are included after every comment of the reviewer below:
Major revisions
- My suggestion is to provide the rationale to evaluate the cardiovascular risk in patients with PKU in relation to their life expectancy or the prevalence of cardiovascular diseases as compared to people without PKU in the introduction. The gap in knowledge in this area should be identified, which would provide a more solid motivation for this review.
Done in the introduction, third paragraph.
- The systematic review identified adipose tissue proportion and distribution as a cardiovascular risk factor. However, other risk factors that are present in the general population were presented in relation to other inherited metabolic diseases but were not included in the analysis in patients with PKU. The presence of dyslipidemia, high blood pressure, age, sex, and smoking are major risk factors that are used in the SCORE diagrams for the evaluation of cardiovascular risk in people without inherent metabolic diseases. Diabetes and prediabetes are also major cardiovascular risk factors. The presence of these factors in the studies addressing patients with PKU should also be presented. If possible, it would be useful to provide a possible explanation for the reason why these factors were not evaluated in the studies included in the analysis.
We presented examples of inherited metabolic diseases with known cardiovascular risk, to show the absence of this knowledge in other IMDs. Fat mass was evaluated as the morphological part of morphofunctional assessment. The presence of major cardiovascular risk factors, as the functional part of morphofunctional assessment, is intended to be reported separately, as the second step after morphological evaluation (fat mass).
CHANGES DONE AS PROPOSED BY OTHER REVIEWERS OR EDITORIAL.
- Project registered at Open Science Framework: https://osf.io/f2xvn
- Are the conclusions supported by the results?
Conclusions corrected.
- Line 25: The following sentence needs revision for proper English language: “Some limitations were not including a healthy control group, nor reporting sex-specific results, and different techniques to assess fat mass.”
Corrected.
- I suggest enlarging the margins of Table 1 to span the entire page, in order to improve its quality.
Done as suggested.
- Table 1: The sex parameter (F/M) should be placed in a separate column.
Done as suggested.
- Line 165: I suggest maintaining the same verb tense (e.g., simple past) throughout the document and avoiding the use of the present simple.
Corrected throughout the whole manuscript.
- Lines 251-258: This paragraph is too long. It is advisable to break it up into smaller parts for clarity.
Done as suggested.
- Lines 412 and 419: Please replace “…” with “etc.”
Corrected.
- The discussion of the results needs significant improvement to avoid merely describing the results obtained.
Discussion enlarged explaining results.
- The manuscript uses overly informal language throughout (e.g., the use of “we” should be avoided). Please revise it.
Revised and corrected English language in the whole manuscript, avoiding first person and active voice, changing it into passive voice.
- In general, the manuscript requires an extensive revision of the English language.
Revised and corrected English language in the whole manuscript.
- The authors should emphasize the importance of experimental confirmation for bioinformatics predictions. How can these predictions be validated using experimental evidence from cell models or patient samples?
New paragraph at the end of the subsection 4.5. Strengths and limitations of this study.
- Examining potential biases in datasets, such as age, gender, and ethnicity distribution, can improve the review's conclusions.
Third and four paragraph in the subsection 4.5. Strengths and limitations of this study.
- The text could benefit by discussing the potential therapeutic uses of the findings. For example, how could these findings be leveraged to create new treatment targets or biomarkers for the early detection of cardiometabolic risk in PKU patients?
Included in the last paragraph of subsection 4.4
- Why include research with both pediatric and adult populations? How were the results adjusted?
New paragraph added in Discusion, 6th paragraph of subsection 4.4. Summary of evidence, and third paragraph counting from the end of subsection 4.5. Strengths and limitations of this study.
- How were differences among reviewers managed during data extraction and bias assessment? Were there any significant disagreements that influenced study selection?
Explained in “2.2. Search Strategy, Study Selection, and Data Collecction”, third paragraph.
- How did the authors adjust for the varying accuracy and precision of fat mass assessment techniques (BIA, DXA, plethysmography, deuterium) in the synthesis of results?
It was not adjusted, differences between groups from every study were presented as it was reported.
Commented in the third paragraph counting from the end of subsection 4.5. Strengths and limitations of this study.
- The manuscript cites greater fat mass in several categories but lacks specific statistical analyses. Can you provide more particular statistical results or p-values to support these trends?
Tables 2 and 3 have been added, showing fat mass results and statistical analyses, when available.
- The review included only one longitudinal study. How do the results of this study relate to cross-sectional studies? What implications do these longitudinal findings have for understanding how fat mass fluctuates over time?
New paragraph added in Discusion, 5th paragraph of subsection 4.4. Summary of evidence.
- The review emphasizes the link between metabolic regulation and fat mass, but acknowledges the lack of reliable data. Can you elaborate on how metabolic control was assessed and how it affected the study's results?
Included at the end of subsecton 4.4. Summary of evidence
Reviewer 2 Report
Comments and Suggestions for Authors
I have reviewed the manuscript titled "Morphofunctional assessment beyond malnutrition: fat mass assessment in adult patients with phenylketonuria. Systematic review," which requires an extensive revision, particularly in terms of the overall organization and the discussion of the results obtained.
- Line 25: The following sentence needs revision for proper English language: “Some limitations were not including a healthy control group, nor reporting sex-specific results, and different techniques to assess fat mass.”
- It is recommended to expand the “Introduction” section, as it lacks an adequate description of the current state of the art, the characteristics of the pathology, and the hypothetical association between PKU and fat mass.
- I suggest enlarging the margins of Table 1 to span the entire page, in order to improve its quality.
- Table 1: The sex parameter (F/M) should be placed in a separate column.
- Line 165: I suggest maintaining the same verb tense (e.g., simple past) throughout the document and avoiding the use of the present simple.
- The work appears somewhat disorganized, especially in the “Discussion” section. It is recommended to follow the MDPI guidelines for the proper subdivision of paragraphs.
- Lines 251-258: This paragraph is too long. It is advisable to break it up into smaller parts for clarity.
- Lines 412 and 419: Please replace “…” with “etc.”
- The discussion of the results needs significant improvement to avoid merely describing the results obtained.
- The manuscript uses overly informal language throughout (e.g., the use of “we” should be avoided). Please revise it.
- In general, the manuscript requires an extensive revision of the English language.
Comments on the Quality of English Language
Extensive editing of Enghlish language required
Author Response
Answers are included after every comment of the reviewer below:
I have reviewed the manuscript titled "Morphofunctional assessment beyond malnutrition: fat mass assessment in adult patients with phenylketonuria. Systematic review," which requires an extensive revision, particularly in terms of the overall organization and the discussion of the results obtained.
- Line 25: The following sentence needs revision for proper English language: “Some limitations were not including a healthy control group, nor reporting sex-specific results, and different techniques to assess fat mass.”
Corrected.
- It is recommended to expand the “Introduction” section, as it lacks an adequate description of the current state of the art, the characteristics of the pathology, and the hypothetical association between PKU and fat mass.
Not expanded as suggested, as descriptions asked for have been already revised recently in reference [10], included in the Introduction.
- I suggest enlarging the margins of Table 1 to span the entire page, in order to improve its quality.
Done as suggested.
- Table 1: The sex parameter (F/M) should be placed in a separate column.
Done as suggested.
- Line 165: I suggest maintaining the same verb tense (e.g., simple past) throughout the document and avoiding the use of the present simple.
Corrected throughout the whole manuscript.
- The work appears somewhat disorganized, especially in the “Discussion” section. It is recommended to follow the MDPI guidelines for the proper subdivision of paragraphs.
Nutrients Microsoft word template file from MDPI “Instructions for authors” (https://www.mdpi.com/journal/nutrients/instructions) has been employed (https://www.mdpi.com/files/word-templates/nutrients-template.dot). Subsections numbered according to template indications, with a maximum 3-level numeration (e.g. 3.4.3. Metabolic control). Paragraphs margins and position according to Nutrients template.
As explained previously, Introduction section has not been expanded. As the main topic of this Special Issue is “Morphofunctional assessment”, and is the only article regarding inherited metabolic diseases in the issue, this is needed to be treated properly. If included in Introduction, this would be too long. This is the reason why those topics are outlined in the Introduction, but explained at the beginning of the Discussion.
- Lines 251-258: This paragraph is too long. It is advisable to break it up into smaller parts for clarity.
Done as suggested.
- Lines 412 and 419: Please replace “…” with “etc.”
Corrected.
- The discussion of the results needs significant improvement to avoid merely describing the results obtained.
Discussion enlarged explaining results.
- The manuscript uses overly informal language throughout (e.g., the use of “we” should be avoided). Please revise it.
Revised and corrected English language in the whole manuscript, avoiding first person and active voice, changing it into passive voice.
- In general, the manuscript requires an extensive revision of the English language.
Revised and corrected English language in the whole manuscript.
CHANGES DONE AS PROPOSED BY OTHER REVIEWERS OR EDITORIAL.
- Project registered at Open Science Framework: https://osf.io/f2xvn
- Are the conclusions supported by the results?
Conclusions corrected.
- Provide the rationale to evaluate the cardiovascular risk in patients with PKU in relation to their life expectancy or the prevalence of cardiovascular diseases as compared to people without PKU in the introduction. The gap in knowledge in this area should be identified, which would provide a more solid motivation for this review.
Done in the introduction, third paragraph
- The authors should emphasize the importance of experimental confirmation for bioinformatics predictions. How can these predictions be validated using experimental evidence from cell models or patient samples?
New paragraph at the end of the subsection 4.5. Strengths and limitations of this study.
- Examining potential biases in datasets, such as age, gender, and ethnicity distribution, can improve the review's conclusions.
Third and four paragraph in the subsection 4.5. Strengths and limitations of this study.
- The text could benefit by discussing the potential therapeutic uses of the findings. For example, how could these findings be leveraged to create new treatment targets or biomarkers for the early detection of cardiometabolic risk in PKU patients?
Included in the last paragraph of subsection 4.4
- Why include research with both pediatric and adult populations? How were the results adjusted?
New paragraph added in Discusion, 6th paragraph of subsection 4.4. Summary of evidence, and third paragraph counting from the end of subsection 4.5. Strengths and limitations of this study.
- How were differences among reviewers managed during data extraction and bias assessment? Were there any significant disagreements that influenced study selection?
Explained in “2.2. Search Strategy, Study Selection, and Data Collecction”, third paragraph.
- How did the authors adjust for the varying accuracy and precision of fat mass assessment techniques (BIA, DXA, plethysmography, deuterium) in the synthesis of results?
It was not adjusted, differences between groups from every study were presented as it was reported.
Commented in the third paragraph counting from the end of subsection 4.5. Strengths and limitations of this study.
- The manuscript cites greater fat mass in several categories but lacks specific statistical analyses. Can you provide more particular statistical results or p-values to support these trends?
Tables 2 and 3 have been added, showing fat mass results and statistical analyses, when available.
- The review included only one longitudinal study. How do the results of this study relate to cross-sectional studies? What implications do these longitudinal findings have for understanding how fat mass fluctuates over time?
New paragraph added in Discusion, 5th paragraph of subsection 4.4. Summary of evidence.
- The review emphasizes the link between metabolic regulation and fat mass, but acknowledges the lack of reliable data. Can you elaborate on how metabolic control was assessed and how it affected the study's results?
Included at the end of subsecton 4.4. Summary of evidence
Reviewer 3 Report
Comments and Suggestions for Authors
Reviewed Manuscript: "Morphofunctional assessment beyond malnutrition: fat mass assessment in adult patients with phenylketonuria. Systematic review."
Strengths:
· The study provides a systematic review of fat mass assessment in adult patients with phenylketonuria (PKU), emphasizing the significance of fat mass in assessing cardiometabolic risk.
· The methodology is well-defined, with clear inclusion and exclusion criteria, a precise search strategy, and a thorough assessment of bias using the NIH Quality Assessment Tool.
· The review contains studies from several geographical regions, increasing the generalizability of the findings.
· The study addresses limitations of the included research, including variation in fat mass assessment procedures and potential bias.
Recommendations:
- The authors should emphasize the importance of experimental confirmation for bioinformatics predictions. How can these predictions be validated using experimental evidence from cell models or patient samples?
- Examining potential biases in datasets, such as age, gender, and ethnicity distribution, can improve the review's conclusions.
- The text could benefit by discussing the potential therapeutic uses of the findings. For example, how could these findings be leveraged to create new treatment targets or biomarkers for the early detection of cardiometabolic risk in PKU patients?
- Why include research with both pediatric and adult populations? How were the results adjusted?
- How were differences among reviewers managed during data extraction and bias assessment? Were there any significant disagreements that influenced study selection?
- How did the authors adjust for the varying accuracy and precision of fat mass assessment techniques (BIA, DXA, plethysmography, deuterium) in the synthesis of results?
- The manuscript cites greater fat mass in several categories but lacks specific statistical analyses. Can you provide more particular statistical results or p-values to support these trends?
- The review included only one longitudinal study. How do the results of this study relate to cross-sectional studies? What implications do these longitudinal findings have for understanding how fat mass fluctuates over time?
- The review emphasizes the link between metabolic regulation and fat mass, but acknowledges the lack of reliable data. Can you elaborate on how metabolic control was assessed and how it affected the study's results?
Overall Assessment:
The study highlights a possible connection between fat mass and cardiometabolic risk and offers insightful information about how fat mass is assessed in adult PKU patients. The review's robustness is increased by the inclusion of a wide range of research, but experimental validation, better figure quality, a more thorough investigation of biases, and a discussion of the clinical implications might all greatly increase the impact and application of the findings. The text would be strengthened even further by the addition of more focused statistical analysis and a thorough investigation of the effects of metabolic control.
Author Response
Authors would like to express our thankfulness for your recommendations to improve the manuscript.
Answers are included after every recommendation of the reviewer below:
Recommendations:
- The authors should emphasize the importance of experimental confirmation for bioinformatics predictions. How can these predictions be validated using experimental evidence from cell models or patient samples?
New paragraph at the end of the subsection 4.5. Strengths and limitations of this study.
- Examining potential biases in datasets, such as age, gender, and ethnicity distribution, can improve the review's conclusions.
Third and four paragraph in the subsection 4.5. Strengths and limitations of this study.
- The text could benefit by discussing the potential therapeutic uses of the findings. For example, how could these findings be leveraged to create new treatment targets or biomarkers for the early detection of cardiometabolic risk in PKU patients?
Included in the last paragraph of subsection 4.4.
- Why include research with both pediatric and adult populations? How were the results adjusted?
New paragraph added in Discusion, 6th paragraph of subsection 4.4. Summary of evidence, and third paragraph counting from the end of subsection 4.5. Strengths and limitations of this study.
- How were differences among reviewers managed during data extraction and bias assessment? Were there any significant disagreements that influenced study selection?
Explained in “2.2. Search Strategy, Study Selection, and Data Collecction”, third paragraph.
- How did the authors adjust for the varying accuracy and precision of fat mass assessment techniques (BIA, DXA, plethysmography, deuterium) in the synthesis of results?
It was not adjusted, differences between groups from every study were presented as they were reported.
Commented in the third paragraph counting from the end of subsection 4.5. Strengths and limitations of this study.
- The manuscript cites greater fat mass in several categories but lacks specific statistical analyses. Can you provide more particular statistical results or p-values to support these trends?
Tables 2 and 3 have been added, showing fat mass results and statistical analyses, when available.
- The review included only one longitudinal study. How do the results of this study relate to cross-sectional studies? What implications do these longitudinal findings have for understanding how fat mass fluctuates over time?
New paragraph added in Discusion, 5th paragraph of subsection 4.4. Summary of evidence.
- The review emphasizes the link between metabolic regulation and fat mass, but acknowledges the lack of reliable data. Can you elaborate on how metabolic control was assessed and how it affected the study's results?
Included at the end of subsecton 4.4. Summary of evidence
CHANGES DONE AS PROPOSED BY OTHER REVIEWERS OR EDITORIAL.
- Project registered at Open Science Framework: https://osf.io/f2xvn
- Are the conclusions supported by the results?
Conclusions corrected.
- Provide the rationale to evaluate the cardiovascular risk in patients with PKU in relation to their life expectancy or the prevalence of cardiovascular diseases as compared to people without PKU in the introduction. The gap in knowledge in this area should be identified, which would provide a more solid motivation for this review.
Done in the introduction, third paragraph
- Line 25: The following sentence needs revision for proper English language: “Some limitations were not including a healthy control group, nor reporting sex-specific results, and different techniques to assess fat mass.”
Corrected.
- I suggest enlarging the margins of Table 1 to span the entire page, in order to improve its quality.
Done as suggested.
- Table 1: The sex parameter (F/M) should be placed in a separate column.
Done as suggested.
- Line 165: I suggest maintaining the same verb tense (e.g., simple past) throughout the document and avoiding the use of the present simple.
Corrected throughout the whole manuscript.
- Lines 251-258: This paragraph is too long. It is advisable to break it up into smaller parts for clarity.
Done as suggested.
- Lines 412 and 419: Please replace “…” with “etc.”
Corrected.
- The discussion of the results needs significant improvement to avoid merely describing the results obtained.
Discussion enlarged explaining results.
- The manuscript uses overly informal language throughout (e.g., the use of “we” should be avoided). Please revise it.
Revised and corrected English language in the whole manuscript, avoiding first person and active voice, changing it into passive voice.
- In general, the manuscript requires an extensive revision of the English language.
Revised and corrected English language in the whole manuscript.
Round 2
Reviewer 2 Report
Comments and Suggestions for Authors
I have reviewed the revisions made by the authors in response to the comments provided. The authors have made the necessary changes to improve the quality of the work that know is suitable for acceptance in its current form.